# Characteristics of Thermosetting Polymer Nanocomposites: Siloxane-Imide-Containing Benzoxazine with Silsesquioxane Epoxy Resins

**DOI:** 10.3390/polym12112510

**Published:** 2020-10-28

**Authors:** Chih-Hao Lin, Wen-Bin Chen, Wha-Tzong Whang, Chun-Hua Chen

**Affiliations:** 1Department of Materials Science and Engineering, National Chiao Tung University, Hsinchu 300093, Taiwan; howardlin@itri.org.tw (C.-H.L.); wtwhang@mail.nctu.edu.tw (W.-T.W.); 2Material and Chemical Research Laboratories, Industrial Technology Research Institute, Chutung, Hsinchu 31040, Taiwan; abin@itri.org.tw

**Keywords:** polysiloxane-imide-containing benzoxazine, polyhedral oligomeric silsesquioxane epoxy, double-decker silsesquioxane epoxy, polymer nanocomposite

## Abstract

A series of innovative thermosetting polymer nanocomposites comprising of polysiloxane-imide-containing benzoxazine (PSiBZ) as the matrix and double-decker silsesquioxane (DDSQ) epoxy or polyhedral oligomeric silsesquioxane (POSS) epoxy were prepared for improving thermosetting performance. Thermomechanical and dynamic mechanical characterizations indicated that both DDSQ and POSS could effectively lower the coefficient of thermal expansion by up to approximately 34% and considerably increase the storage modulus (up to 183%). Therefore, DDSQ and POSS are promising materials for low-stress encapsulation for electronic packaging applications.

## 1. Introduction

Compared with pristine polymer nanocomposites, hybrid organic–inorganic nanocomposites comprising of functional polymers as the matrix and nanoscale inorganic constituents have attracted greater interest in both academia and industry because of their tunable and generally more favorable thermal, mechanical, electrical, and barrier properties [1,2,3]. Upgrading current thermosetting polymers has become critical because of their utilization in various applications. Such as fuel cells [4], catalysis [5], membranes [6], CO_2_ adsorption [7], metal uptake [8], coatings [9], thermal insulation [10], and super capacitor [11]. Studies using nanoparticles for modifying thermosetting polymers have revealed the feasibility of hybrid strategies [12,13,14].

Polybenzoxazine (PBZ) is considered a high-performance thermosetting resin because of its excellent thermal stability, high modulus, low water absorption, low surface free energy, and near-zero shrinkage upon curing [15,16,17]. Those properties are superior to some other thermosetting polymers. Crucially, PBZ requires simple procedures of thermal activated ring opening polymerization without the need for catalysts, and do not form byproducts. The low surface free energy property would allow it to have many potential applications as mold release material, lithographic patterning, and super hydrophobic surface material [18,19,20,21,22,23,24]. However, high curing temperatures and poor mechanical properties (brittleness) severely limit PBZ applications. Modification of benzoxazine monomer structures, blending with other polymers, and addition of nanoscale fillers or fibers are the currently major approaches to overcoming the drawbacks of PBZ [25,26,27,28]. Chen et al. demonstrated that siloxane-imine-containing benzoxazine (SiBZ) monomers [29,30] could be used to synthesize PBZ with improved flexibility and weather resistance and reduced surface free energy. However, the conjunction of the siloxane segment and benzoxazine can directly increase the coefficient of thermal expansion (CTE) and thus pose a problem of dimensional instability.

Silsesquioxane, with a general formula of (RSiO_3/2_)_n_ is a representative organic–inorganic compound. Cubic polyhedral oligomeric silsesquioxane (POSS), a silsesquioxane derivative, consists of an inorganic silsesquioxane core with a pore diameter of 0.3–0.4 nm and organic functional groups at the vertex of the cubic silsesquioxane frame. The introduction of organic substituents, such as alkyl, aryl, epoxy, or amine groups, into POSS has improved compatibility between the POSS molecule and polymer matrices [31,32,33,34,35]. Another silsesquioxane derivative, double-decker silsesquioxane (DDSQ) contains two substituents, such as epoxy or allyl groups, in diagonal positions [36,37,38,39]. Incorporation of POSS or DDSQ into polymer matrices is an effective strategy to enhance thermal stability and mechanical and dielectric properties [40,41].

Regarding the enhancement of the dimensional stability, although several studies have introduced cubic POSS to various polymer matrices, the resulting CTEs are not satisfactory. For instance, the use of POSS considerably reduced the CTE of a polyimide (PI) matrix [42]. By contrast, the presence of the octa-functionalized POSS epoxy in a thermosetting epoxy/HHPA matrix exhibited less influence on the CTE, which was attributed to the flexible organic tethers around the POSS cage forming a soft interphase [43].

Previous research also incorporate POSS or DDSQ with conventional PBZ and demonstrate the enhancement of thermal oxidation stability, mechanical properties [44,45,46]; however, a high rigid structure of PBZ, POSS, or DDSQ make the nanocomposites become brittle. In this study, we focused on the improvement of the dimensional stability of more flexible polysiloxane-imide-containing benzoxazine (PSiBZ). Two potential candidates, namely, octa-functionalized POSS epoxy and 3,13-diglycidyloxypropyloctaphenyl DDSQ epoxy, were selected as the organic–inorganic hybrid dopants for the PSiBZ matrix because of their caged silsesquioxane structures and chemical affinities with the matrix. In the molecular structure, DDSQ has larger cage dimensions than cubic POSS and thus is expected to provide desired thermal dimension stability (reduced CTEs) for organic–inorganic nanocomposite materials. The design concept is shown in Scheme 1.

## 2. Materials and Methods

### 2.1. Materials

Phenyltrimethoxysilane (97%), dichloromethylsilane (97%), triethylamine (99.5%), isopropyl alcohol (99.5%), and allyl glycidyl ether (99%) were purchased from Sigma-Aldrich (St. Louis, MO, USA). Sodium hydroxide was purchased from UniRegion Bio-Tech (Hsinchu, Taiwan). Tetrahydrofuran (95%) was purchased from Fisher Chemical (Pittsburgh, PA, USA). Toluene (99.5%) was obtained from J.T. Baker (Phillipsburg, NJ, USA). Tris(dibutylsulfide)rhodium trichloride was obtained from Gelest (Morrisville, PA, USA). N,N-dioctadecylmethylamine was supplied by Fluka (Tokyo, Japan). Activated charcoal powder was obtained from Showa (Saitama, Japan). Siloxane-imide-containing benzoxazine monomer (SiBZ) was provided by Industrial Technology Research Institute (Hsinchu, Taiwan). Octa-functionalized POSS epoxy (70 wt%) in cyclopentanone was purchased from Hybrid Plastics (Hattiesburg, MS, USA). DDSQ epoxy was prepared with reference to previous literature [38,40]. The following is the structure of the as-synthesized DDSQ epoxy was confirmed through ^1^H nuclear magnetic resonance (NMR): 0.31 (s, Si-CH_3_), 0.74 (t, Si-CH_2_-), 1.70 (m, Si-CH_2_-CH_2_-), 2.45 and 2.67 (m, -O-CH_2_-, epoxide), 2.98 (m, -O-CH-, epoxide), 3.17 and 3.45 (m, Si-(CH_2_)_3_-O-CH_2_-], and 3.34 ppm (m, Si-CH_2_-CH_2_-CH_2_-O-). The chemical structures of the SiBZ monomer, DDSQ epoxy, and POSS epoxy are displayed in Figure 1a–c, respectively.

### 2.2. Preparation of DDSQ Epoxy/PSiBZ and POSS Epoxy/PSiBZ Nanocomposites

The weight ratios of the prepared DDSQ epoxy/PSiBZ and POSS epoxy/PSiBZ nanocomposites are displayed in Table 1. A mixture of the POSS or DDSQ epoxy and SiBZ monomers was dissolved in toluene to form a homogeneous solution with a solid content of 50 wt%. The mixture was then poured into an aluminum pan to remove the solvent through heating at 100 °C for 5 h. According to the results of differential scanning calorimetry (DSC) analysis, pure SiBZ and DDSQ epoxy/SiBZ were cured at 200 and 230 °C for 2 h. POSS epoxy/SiBZ were cured at 200 °C for 2 h and 230 °C for 6 h.

### 2.3. Instrumentation

Proton nuclear magnetic resonance (^1^H-NMR) analysis was performed using an NMR spectrometer (Varian 500 MHz; Agilent, Santa Clara, CA, USA) with CDCl_3_ as the solvent and tetramethylsilane (TMS) as a reference. Differential scanning calorimetry (DSC Q10; TA Instruments, New Castle, DE, USA) was conducted under a nitrogen atmosphere. Approximately 10 mg of the testing sample was sealed in an aluminum pan. The samples were heated from 40 to 250 °C at a heating rate of 10 °C/min. The onset temperature, peak temperature, and enthalpy of the exothermic peak of the cured compositions were recorded. Thermal degradation properties were measured using a thermogravimetric analyzer (TGA, Q500; TA Instruments, New Castle, DE, USA) from 30 to 800 °C at a heating rate of 10 °C/min. Dynamic mechanical analysis (DMA, Q800; TA Instruments, New Castle, DE, USA) was performed at a heating rate of 5 °C/min from −80 to 280 °C with a fixed frequency of 1 Hz in the single cantilever mode. The dimensions of the test species were 17.5 mm (L) × 10.0 mm (W) × 0.5 mm (T). Thermomechanical analysis (TMA, Q400; TA Instruments, New Castle, DE, USA) was conducted at a heating rate of 10 °C/min from 30 to 250 °C, and a force of 0.05 N was applied.

## 3. Results and Discussion

### 3.1. Curing Behavior of DDSQ Epoxy/PSiBZ and POSS Epoxy/PSiBZ Nanocomposites

The curing reactions of the pure PSiBZ, DDSQ epoxy/PSiBZ nanocomposites and POSS epoxy/PSiBZ nanocomposites are illustrated in Scheme 2. The SiBZ monomer can self-cure through thermal activated ring-opening polymerization and co-cure with DDSQ or POSS epoxy to form DDSQ epoxy/PSiBZ and POSS epoxy/PSiBZ thermosetting nanocomposites under a high temperature condition [47].

The DSC thermograms of the pure PSiBZ, the DDSQ epoxy/PSiBZ, and POSS epoxy/PSiBZ nanocomposites are shown the exothermic curve and displayed in Figure 2 and the summary of the curing characteristics are listed in Table 2. In the DDSQ epoxy/PSiBZ system, both the exothermic onset and peak temperatures increased with the percentage by weight of DDSQ epoxy. A similar tendency was observed for POSS epoxy/PSiBZ; however, the steric hindrance of the eight epoxy group in the vicinity position of the POSS core was higher than two epoxy group in diagonal position of DDSQ core, which resulted in the exothermic onset and exothermic with a maximum temperature of POSS epoxy/PSiBZ higher than those of DDSQ epoxy/PSiBZ and thus the curing time at 230 °C of POSS epoxy/PSiBZ was set longer than DDSQ epoxy/PSiBZ to ensure the cured reaction completely.

The chain mobility of PSiBZ was restricted by the incorporation of the rigid DDSQ or POSS cage into its structure; therefore, the onset and peak temperatures increased with the DDSQ or POSS content. Liu et al. also found a similar curing phenomenon in a PBZ-DDSQ copolymer. The onset and peak temperatures of the PBZ-DDSQ copolymer increased with the DDSQ content [44].

### 3.2. Thermal Stability of DDSQ Epoxy/PSiBZ and POSS Epoxy/PSiBZ Nanocomposites

The thermal stability of the pure PSiBZ, the DDSQ epoxy/PSiBZ nanocomposites and the POSS epoxy/PSiBZ nanocomposites was analyzed by TGA under air atmosphere as displayed in Figure 3a,b. The found 5% mass loss temperature (*T*_d5%_) was 419 °C for the pure PSiBZ and slightly decreased to 413 °C and 410 °C for 20 wt% and 40 wt% DDSQ epoxy/PSiBZ, respectively. The decrease in *T*_d5%_ should originate from the reacted two epoxy groups in the DDSQ Epoxy/PSiBZ nanocomposites. In the case of the POSS epoxy/PSiBZ nanocomposites, *T*_d5%_ for 20 wt% and 40 wt% POSS epoxy/PSiBZ was 392 °C and 396 °C, respectively, reasonably due to the presence of eight reacted epoxy groups of POSS with PSiBZ, leading to a lower thermal stability than the DDSQ epoxy/PSiBZ nanocomposites. The char yield of the DDSQ epoxy/PSiBZ and POSS epoxy/PSiBZ nanocomposites also increased with the increase of the DDSQ or POSS. As a result, the DDSQ epoxy/PSiBZ nanocomposites exhibited better thermal stability than the POSS epoxy/PSiBZ nanocomposites.

### 3.3. Dynamic Mechanical Properties of DDSQ Epoxy/PSiBZ and POSS Epoxy/PSiBZ Nanocomposites

The dynamic mechanical properties of the DDSQ epoxy/PSiBZ and POSS epoxy/PSiBZ nanocomposites from −100 to 280 °C were characterized through DMA. The storage modulus and tanδ of the DDSQ epoxy/PSiBZ and POSS epoxy/PSiBZ nanocomposites are presented in Figure 4 and Table 3. The storage modulus of pure PSiBZ was 960 MPa at 25 °C. After 20 and 40 wt% DDSQ epoxy was added, the storage modulus increased to 1506 and 1516 MPa, respectively. In the 40 wt% POSS epoxy/PSiBZ, a higher storage modulus of 1753 MPa was obtained. The increases were approximately 158% and 183% for DDSQ and POSS, respectively, indicating superior reinforcement with POSS.

The peak temperature from the tanδ curves of the DDSQ/PSiBZ nanocomposites revealed that the addition of 20 and 40 wt% of DDSQ lowered *T*_g_ from the 157 °C of pure PSiBZ to 155 and 140 °C, respectively. The eight bulky phenyl groups of the DDSQ epoxy disturbed the chain stack to increase larger free volume in the cross-linked structure and thus reduce the *T*_g_ [44]. By contrast, the addition of 20 wt% POSS epoxy resulted in a higher *T*_g_ of 226 °C. Notably, *T*_g_ becomes not obvious when the POSS epoxy content was 40 wt%. The increased *T*_g_ can be attributed to the increased cross-linking caused by the presence of eight epoxy groups of POSS. Furthermore, at 40 wt% POSS epoxy, the apparent nano effect caused by strong interaction between nanoscale POSS epoxy and PSiBZ was found; consequently, *T*_g_ became not obvious. Similar nano effect phenomena have been previously observed [48,49].

There are two major factors that led to the reinforcement of the DDSQ/PSiBZ and POSS/PSiBZ nanocomposites: (i) the rigidity of DDSQ and POSS and (ii) the *T*_g_ (directly related to the crosslinking density) of the DDSQ/PSiBZ and POSS/PSiBZ nanocomposites networks. In the case of DDSQ/PSiBZ nanocomposites, the storage modulus increased with increasing the DDSQ content due to the rigidity of DDSQ; however, the increased DDSQ simultaneously resulted in a lower *T*_g_, which would lead to the decrease of modules because of the decreased crosslinking density. The very close modulus found from the specimen of the 20 or 40 wt% of DDSQ epoxies could thus be considered as the competition of these two factors. On the other hand, in the POSS/PSiBZ system, both of the rigidity and *T*_g_ increase with the increase of the POSS epoxy. The eight epoxy groups in POSS could provide more crosslinking points in the POSS/PSiBZ network. As a consequence, the superior reinforcement effect could be obtained for the POSS/PSiBZ nanocomposites.

### 3.4. Thermomechanical Properties of DDSQ Epoxy/PSiBZ and POSS Epoxy/PSiBZ Nanocomposites

The CTEs of the DDSQ epoxy/PSiBZ and POSS epoxy/PSiBZ nanocomposites were investigated through TMA and compared with that of pure PSiBZ. Figure 5 illustrates the dimensional changes and CTEs of the pure PSiBZ, DDSQ epoxy/PSiBZ nanocomposites, and POSS epoxy/PSiBZ nanocomposites below the glass transition temperature (*T*_g_). Figure 5a revealed that CTE 1 (below *T*_g_ at a temperature range of 40–75 °C) considerably decreased with an increase in DDSQ content. From 0 to 40 wt% of DDSQ, the decreases in CTE 1 were approximately 34%. Notably, unlike the DDSQ epoxy system, the POSS epoxy/PSiBZ nanocomposites show the estimated changes in CTE 1 were approximately 14% (Figure 5b). The present results indicate that as the temperature decreased below *T*_g_, the molecules lose energy under the glassy state, only short-range molecular movement occurred, and the larger core dimension and the phenyl substituents at the vertex of the DDSQ frame efficiently restricted the chain mobility of PSiBZ. Thus, noticeable variation was observed in CTE 1. The results indicate that both adding 40 wt% DDSQ epoxy and POSS epoxy demonstrated the significant CTE 1 reduced the effect of PSiBZ; furthermore, a DDSQ epoxy exhibited more of a CTE 1 reduce effect than the POSS epoxy.

## 4. Conclusions

In this study, DDSQ epoxy and POSS epoxy were added to PSiBZ to achieve superior thermomechanical and dynamic mechanical properties. The addition of 40 wt% DDSQ epoxy resulted in a 34% reduction in CTE 1 and 158% increase in the storage modulus. The introduction of the 40 wt% POSS epoxy increased the storage modulus by 183%. This study not only serves as proof of the design concept but also experimentally demonstrated the potential of the DDSQ and POSS epoxy as functional additives for PSiBZ for lowering the CTE while enhancing the thermomechanical properties.

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
