# Peer review of "Characteristics of Thermosetting Polymer Nanocomposites: Siloxane-Imide-Containing Benzoxazine with Silsesquioxane Epoxy Resins"

_polymers, 2020, doi:10.3390/polym12112510_

Round 1

Reviewer 1 Report

The manuscript entitled ' Characteristics of thermosetting polymer nanocomposites: Siloxane-imide-containing benzoxazine with silesquinoxane epoxy resins' submitted by Lin and co-workers describes the co-polymerization of imide-containing benzoxazine and double-decker or polyhedral silsesquioxane epoxys (DDSQ and POSS) with the aim to improve thermomechanical properties of the polybenzoxazine material.

The authors compared two type of silsesquiloxane in various ratio to the neat polybenzoxazine material to asses their effect on the thermomechanical properties, especially on the thermal expansion of the materials. 

The authors are introducing briefly the concept of thermosets nanocomposites before introducing benzoxazines and silsesquinoxane materials. Nevertheless, the introduction is lacking some references. 

Line 37-39: reference 10 is inappropriate to fully describe the sentence and some additional references should be added here. 

In addition, the authors introduced siloxane-containing benzoxazines (l 39-43) mainly to discuss the contribution regarding thermal expansion of the material. Unfortunately, authors did indtroduce silsesquinoxane alone and with other polymer matrices (epoxy and polyimide), but they did not introduce their use combined with benzoxazines. This last point is regrettable as silsesquinoxane were used in benzoxazines before in different ways. The authors should add a paragraph concerning this specific point and review the different ways silsesquinoxane were incorporated in benzoxazines and address the novelty of their work in the field.

The experiments were conducted in a logical manner. However some changes hereafter described should be done in order to facilitate the comprehension of the manuscript. 

  • l111: 'the SiBZ monomer can be self-cured through ring-opening polymerizarion' the authors should indication under thermal activation
  • Scheme 1: the concept of thermal activation to react the silsesquinoxane and the benzoxazine monomer should appear on the scheme.
  • l126: the authors comment on the activation energies obtained by DSC. The values should be added and discussed. 
  • l137: 'before the glass transition temperature' : the authors are not indicating at this stage of the manuscript the Tg of their materials. The CTE is discussed before the DMA, therefore I would recommend to move the CTE part after the description of DMA. Furthermore, the authors should clearly indicate on which range of temperature the CTE were calulated.
  • l156-159: the authors are assuming based on the DMA results that superior reinforcement by the incorporation of POSS epoxy is due to superior crosslinking density. The authors should support this assumption by calculating the crosslinking density from the DMA is possible or by swelling tests.
  • Interestingly the addition of 20 or 40% of DDSQ leads to similar modulus and similar tangent delta values as it is not the case for 20% and 40 % of POSS where the values are clearly differing. Could the authors add additional comment on this? 
  • l174 scheme3: The temperature at which the modulus values were taken should be indicated. 

As the aim of this work is assess the reinforcement of the thermomechanical properties of a polybenzoxazine matrix it would be interesting to discuss the thermal stability of the materials as well by some TGA analyses. 

The authors should also revise carefully the references where several mistakes were noticed, for examples titles missing, mistakes in authors name spelling (ref 10).

Author Response

The response letter to reviewer is in attach file.

Reviewer 2 Report

The authors just need to check text for small spelling errors such as line 145 CET instead of CTE.

Author Response

(The authors gave the same response as above.)

Reviewer 3 Report

Characteristics of Thermosetting Polymer Nanocomposites: Siloxane-Imide-Containing Benzoxazine with Silsesquioxane Epoxy Resins

 In this manuscript, preparation and properties of Siloxane-Imide-Containing Benzoxazine with Silsesquioxane Epoxy Resins were investigated. The manuscript needs improvement and a comprehensive introduction to polybenzoxazine should be given. Therefore, the manuscript is not yet ready to be published in this journal. I recommended the authors address adequately the following points:

  1. Lines 19-21: “Therefore, DDSQ and POSS are promising materials for low-stress coating or electronic packaging applications.” My concern for “low-stress coating” for this application, contact angle results should be provided.
  2. Line 29 1nd 30: “Upgrading current thermosetting polymers has become critical because of their utilization in various applications” The following sentence should be added: “such as fuel cells [Macromolecules2012, 45, 3, 1438–1446, https://doi.org/10.1021/ma202694p], catalysis [Green , 2020,22, 1209-1219, https://doi.org/10.1039/C9GC03504D], membranes [Desalination 267(1):73-81, DOI: 10.1016/j.desal.2010.09.008], CO2 adsorption [ACS Sustainable Chem. Eng. 2016, 4, 3, 1286–1295, https://doi.org/10.1021/acssuschemeng.5b01323], metal uptake [Desalination 256(1):108-114, DOI: 10.1016/j.desal.2010.02.005], Coatings [RSC Adv., 2016,6, 28428-28434, https://doi.org/10.1039/C6RA02215D], thermal insulation [Macromolecular Sci. and Eng. 2019, 304(7), 1900137, https://doi.org/10.1002/mame.201900137], and super capacitor (Mater. Sci. Eng.: B, 2010,167(1), 36-42, https://doi.org/10.1016/j.mseb.2010.01.024”
  3. Lines 33-43: The introduction regarding to polybenzoxazine was short and inefficient. Therefore, it should be extended.
  4. Lines 37-39: “Modification of benzoxazine monomer structures, blending with other polymers, and addition of nanoscale fillers or fibers are the currently major approaches to overcoming the drawbacks of PBZ [10]” The authors should add other studies including (Biomacromolecules2013, 14, 6, 1806–1815, https://doi.org/10.1021/bm4002014). (Polymer, 1996, 37(20), 4487-4495, https://doi.org/10.1016/0032-3861(96)00303-5), (Polymer 52(2):307-317, DOI: 1016/j.polymer.2010.11.052)
  5. Lines 80-82: “31 (s, Si-CH3), 0.74 (t, Si-80 CH2-), 1.70 (m, Si-CH2-CH2-), 2.45 and 2.67 (m, -O-CH2-, epoxide), 2.98 (m, -O-CH-, epoxide), 3.17 81 and 3.45 (m, Si-(CH2)3-O-CH2-], and 3.34 ppm (m, Si-CH2-CH2-CH2-O-).” This part should be revised. The number of the atoms should be written in subscript.
  6. Lines 92-94: “Pure SiBZ and DDSQ epoxy/SiBZ were cured at 200 and 230 °C for 2 h. POSS epoxy/SiBZ were cured at 200 °C for 2 h and 230 °C for 6 h.” Why the authors did not cure the both composites with the same curing procedure (i.e. heating 2 h for each temperature)?
  7. Lines 116 and 117: “The DSC exothermic curves and curing characteristics of the DDSQ epoxy/PSiBZ and POSS epoxy/PSiBZ nanocomposites are displayed in Figure 2 and Table 2.” The sentence should be revised to read “The DSC thermograms of the DDSQ epoxy/PSiBZ and POSS epoxy/PSiBZ nanocomposites are displayed in Figure 2 and the summary of the curing characteristics are listed in Table 2.”
  8. Line 118: I recommend to change “peak temperature to “exothermic with a maximum temperature”
  9. Lines 119-123:“the steric hindrance of the cyclic epoxy group in the POSS epoxy results in exothermic onset and peak temperatures of POSS epoxy/PSiBZ higher than those of DDSQ epoxy/PSiBZ and thus the curing time at 230oC of POSS epoxy/PSiBZ is setted longer than DDSQ epoxy/PSiBZ to ensure the cured reaction completely.” This sentence should be revised.
  10. Line 125: The word “discovered” should be replaced by “found”
  11. Table 2: According to my comment number 8, “Tpeak” should be changed to “Tmax
  12. Line 144-146: “The results indicate that both DDSQ epoxy and POSS epoxy demonstrate the CET 1 reduce effect of PSiBZ; furthermore, DDSQ epoxy exhibits more CTE 1 reduce effect than POSS epoxy.” The sentence should be corrected.
  13. Figure 3. I recommended this Figure to be combined as: The revised Figure 3a illustrates the data that represents now in Figures 3(a) and (c). The revised Figure 3b displays the data that represents now in Figures 3(b) and (b).
  14. Line 156: “Mpa” should be written correctly as “MPa”
  15. Line 157: “Approximately 58% and 82%” both numbers “58” and “82” should be rechecked.
  16. Line 161: “Tg” should be corrected. The letter “g” should be subscript.

Author Response

The response letter to reviewer is in attach file

Reviewer 4 Report

Systematic investigation of PSiBZ/DDSQ epoxy and PSiBZ/POSS epoxy composites were explored, showing improved thermomechanical and dynamic mechanical properties compare to that of pristine PSiBZ. Although the authors clearly presented the thermal and mechanical properties of the polymers, there are no deep explanation of why such polymer composites can achieve improved performance. The authors should include the morphological characteristics of the studied polymers to understand why the thermomechanical and dynamic mechanical properties of PSiBZ can be enhanced after adding DDSQ or POSS epoxy. AFM and TEM characterizations are highly suggested to be included in the manuscript. A major revision is required before publication.

Author Response

The response letter to reviewer is in the attach file.

Round 2

Reviewer 1 Report

The authors made the necessary changes and improved the quality of the manuscript. The paper is now ready for a possible publication in Polymers. 

Reviewer 4 Report

The authors improved the quality of manuscript after revision. I recommend it to be published on Polymers